# The Vitamin C Enantiomers Possess a Comparable Potency in the Induction of Oxidative Stress in Cancer Cells but Differ in Their Toxicity

**DOI:** 10.3390/ijms25052531

**Published:** 2024-02-22

**Authors:** Dinara Begimbetova, Agata N. Burska, Aidana Baltabekova, Assiya Kussainova, Assiya Kukanova, Fatima Fazyl, Milana Ibragimova, Kenzhekyz Manekenova, Abay Makishev, Rakhmetkazhi I. Bersimbaev, Dos D. Sarbassov

**Affiliations:** 1National Laboratory Astana, Nazarbayev University, Astana 010000, Kazakhstan; dinara.begimbetova@nu.edu.kz (D.B.); agata.burska@nu.edu.kz (A.N.B.); assya.kussainova@gmail.com (A.K.); assiya.kukanova@gmail.com (A.K.); fatima.fazyl@nu.edu.kz (F.F.); milanaibragimova2602@yandex.ru (M.I.); 2School of Medicine, Nazarbayev University, Astana 010000, Kazakhstan; aidana.baltabekova@nu.edu.kz; 3Department of Experimental Medicine, University of Genoa, 16132 Genoa, Italy; 4Department of General Biology and Genomics, Institute of Cell Biology and Biotechnology, L.N. Gumilyov Eurasian National University, Astana 010008, Kazakhstan; ribers@mail.ru; 5Department of Oncology, Astana Medical University, Astana 010000, Kazakhstan; makishev.a@amu.kz; 6Department of Pathological Anatomy, Astana Medical University, Astana 010000, Kazakhstan; kena_31@mail.ru; 7Department of Biology, School of Sciences and Humanities, Nazarbayev University, Astana 010000, Kazakhstan

**Keywords:** vitamin C (VC) or ascorbic acid, D-Isoascorbic acid (D-VC), L-ascorbic acid (L-VC), toxicity, reactive oxidative species (ROS)

## Abstract

The use of vitamin C (VC) in high doses demonstrates a potent tumor suppressive effect by mediating a glucose-dependent oxidative stress in Kirsten rat sarcoma (KRAS) mutant cancer cells. VC with arsenic trioxide (ATO) is a promising drug combination that might lead to the development of effective cancer therapeutics. Considering that a tumor suppressive effect of VC requires its high-dose administration, it is of interest to examine the toxicity of two enantiomers of VC (enantiomer d-optical isomer D-VC and natural l-optical isomer L-VC) in vitro and in vivo. We show that the combinations of L-VC with ATO and D-VC with ATO induced a similar cytotoxic oxidative stress in Kras^G12D^-expressing mutant cancer cells as indicated by a substantial increase in reactive oxidative species (ROS) production and depolarization of mitochondria. To examine the L-VC and D-VC toxicity effects, we administered high doses of D-VC and L-VC to CD1 mice and carried out an evaluation of their toxic effects. The daily injections of L-VC at a dose of 9.2 g/kg for 18 days were lethal to mice, while 80% of mice remained alive following the similar high-dose administration of D-VC. Following the drug injection courses and histopathological studies, we determined that a natural form of VC (L-VC) is more harmful and toxic to mice when compared to the effects caused by the similar doses of D-VC. Thus, our study indicates that the two enantiomers of VC have a similar potency in the induction of oxidative stress in cancer cells, but D-VC has a distinctive lower toxicity in mice compared to L-VC. While the mechanism of a distinctive toxicity between D-VC and L-VC is yet to be defined, our finding marks D-VC as a more preferable option compared to its natural enantiomer L-VC in clinical settings.

## 1. Introduction

The oncogenic KRAS mutations are the critical drivers of tumorigenesis in a variety of types of human cancers. They occur in almost all (95%) pancreatic ductal adenocarcinomas (PDACs) [1] and are common in colorectal (about 40%), lung (20%) and other types of cancers [2]. KRAS is a member of the small guanosine triphosphatase GTPase family engaged in growth factor signaling by coupling the activated receptor tyrosine kinase to the mitogen-activated protein kinases, extracellular signal-regulated kinases (RAF/MEK/MAPK) and the phosphatidylinositol 3′–kinase (PI3K) signaling pathways. A GTPase activity of KRAS regulates cell proliferation and survival [3]. Deregulation of KRAS by enhancing its GTP binding affinity leads to the ligand-independent hyperactivation of growth factor signaling that leads to uncontrolled cell proliferation and tumorigenesis [4].

KRAS mutant cancers represent highly malignant human cancers with poor outcomes. KRAS is a small G protein and its direct targeting remains a highly challenging task [5]. Although targeted therapy for some cancers has had some success, patients with pancreatic and colorectal cancers have responded poorly to the treatments [6]. Approaches to suppress KRAS by inhibiting its downstream signaling pathways by directly targeting its downstream effectors [7] and the regulation of epigenetic mechanisms [8,9] were not effective. The recent advances in targeting specific mutations of KRAS were highly effective and promising in suppressing the KRAS mutant-driven tumorigenesis but the drug resistance was acquired in a short period of time by selecting the additional mutants of KRAS interfering with the drug binding. Therefore, the search for new methods and strategies for the treatment of KRAS mutant cancers is a critical area in oncology.

Vitamin C (VC), also known as ascorbic acid, has been a subject of interest in the context of cancer treatment [10,11]. VC is an essential nutrient for overall health based on its electron-donating property and it functions as an enzymatic cofactor in a variety of reducing reactions within deoxygenases (involved in the synthesis of collagen and carnitine) and monooxygenases (required for the synthesis of hormones). VC is defined as a critical antioxidant because it is not made in human cells and its deficiency leads to scurvy. Moreover, VC’s functions have been also linked to the fundamental cellular processes including transcription and translation via different mechanisms [10]. Recently, high-dose VC applications became an attractive approach in cancer therapy, especially for KRAS mutant cancers, and it is a topic of ongoing research. In the 1970s, VC at high doses was reported to demonstrate an efficacy in suppressing tumorigenesis [12]. In recent years, more studies have emerged related to the therapeutic potential of VC in pancreatic and colorectal cancers [13,14].

VC shows distinctive effects on primary and cancer cells [15]. In cancer cells, a metabolic shift takes place when cells absorb glucose at a higher rate and use predominantly glycolysis coupled to a diminished rate of oxidative phosphorylation known as the Warburg effect or aerobic glycolysis. The altered cancer metabolism results in an active glucose absorption necessary for accelerated cell growth, which is especially relevant for KRAS mutant cancer cells [4]. It has been reported that VC is oxidized in the body to dehydroascorbate (DHA). DHA in the form of a bicyclic hemiketal structurally resembles glucose and is actively absorbed by cancer cells mediated by a glucose transporter system. Inside a cell, DHA is reduced back to VC by the cost of oxidizing glutathiones (GSHs). In tumor cells, a high level of DHA becomes a substantial burden by depleting cellular GSH and causing an induction of oxidative stress that might lead to a cytotoxic effect [16].

The two enantiomers of VC known as the L- and D-forms of ascorbic acid (D-VC and L-VC) in combination with arsenic trioxide have a similar potent cytotoxic oxidizing effect on cancer cells, but the use of D-VC has been shown to be more effective in suppressing KRAS mutant tumor growth [4,17]. It has been proposed that the rate of oxidation of D-VC to DHA is much slower compared to the oxidation rate of its natural form, L-VC. It might result in a prolonged accumulation of DHA and have a more potent effect on cancer cells.

The purpose of this study was to evaluate the efficacies of L-VC and D-VC in killing Kras^G12D^-expressing cancer cells and also to examine their toxicity in mice.

## 2. Results

### 2.1. The Similar Cytotoxic Effects of D-VC and L-VC in Kras^G12D^-Expressing Cancer Cells

A natural form of VC (L-VC) has been actively studied in the previous studies and only a few reports show a potent cytotoxic effect of L-VC or D-VC in combination with ATO in HCT116 (KRAS mutant) colorectal cancer cells including tumor growth suppression [17]. How D-VC works was not characterized and to compare the action of D-VC with its natural form, L-VC, we examined their cytotoxic effects in the AK192 (Kras^G12D^-inducible mouse pancreatic) cancer cell line [18].

Microscopic analysis showed a strong cytotoxic effect of the combination of ATO and D-VC or L-VC, and each of these compounds alone was not effective in inducing a cytotoxic effect (Figure 1A) that is coherent with our previous study [19]. Here, we show that the combination of L-VC or D-VC with ATO leads to a significant cell death mediated by triggering the generation of reactive oxidative species (ROS) in AK192 cancer cells (Figure 1B) in comparison to untreated control cells (growth medium, GM).

In the control sample, the percentage of cancer cells producing ROS (Mitosox positive cells) was 12.7, the oxidizing compound ATO increased it to 31.4, whereas D-VC or L-VC alone showed the percentage values of 25.9 and 29.2, respectively. Much higher levels of ROS-producing cancer cells were detected following the combination treatments of ATO with L-VC or D-VC that led to 81.8 and 83.3 percent, respectively (in comparison to control *p* < 0.001 for both). The combination of ATO with VC (L-VC or D-VC) induced an ROS generation in cancer cells that was approximately 2.6–3.2 fold higher in comparison to action of ATO, D-VC or L-VC alone. There was no significant difference in ROS generation between the individual drug effects of ATO, L-VC or D-VC, whereas the combination of ATO with L-VC or with D-VC induced a substantial and similar increase in ROS generation, where oxidative stress was detected in more than 80% of cancer cells. This study clearly indicates that D-VC and L-VC have the same potency in the induction of oxidative stress in cancer cells that has been enhanced substantially by another oxidizing compound, ATO.

The mitochondrial membrane potential reflects mitochondrial activity coupled with ATP synthesis. Mitochondria maintain polarized membranes mediating a proper function of the electron transport chain complexes and oxidative phosphorylation, which are essential for energy production in the form of ATP [20]. We examined a functional state of mitochondrial membrane potential in AK192 cancer cells treated with ATO or either a VC enantiomer alone or with their combinations as described in Figure 1. To carry out the mitochondrial functional study, we performed the double staining of mitochondria by MitoTracker Green (MTG) and MitoTracker Deep Red (MTDR). Both stains are mitochondria-selective but differ in staining specificity. MTG reacts with free thiol groups of cysteine and covalently binds to mitochondrial proteins. It accumulates in the mitochondrial matrix. A binding of MTG is independent of mitochondrial membrane potential and is used to measure mitochondrial mass [21]. The second dye, MTDR, is actively pumped inside healthy polarized mitochondria because it is dependent on the mitochondrial membrane potential (MMP) [22].

According to the obtained data, an increased ROS production is associated with a significant depolarization of mitochondria in AK192 cancer cells. A substantial decrease in the polarized mitochondrial membrane (the ratio of MTG high/MTDR low) was detected in AK192 cancer cells treated with the combination of ATO with D-VC or with L-VC in comparison to control cells (at least 4-fold decrease). The detection of a dysfunctional mitochondria (a decrease in the polarized mitochondria membrane) was 2-fold higher in ATO/D-VC- or ATO/L-VC-treated cancer cells compare to the treatment with ATO alone or 3.7-fold higher than in D-VC and 3.2-fold higher than in L-VC-treated cells (Figure 2). Thus, both VC enantiomers possess a similar capacity in the induction of oxidative stress in AK192 cancer cells and their effect is enhanced substantially by the oxidizing compound ATO that leads to a depolarization of the mitochondrial membrane and, subsequently, to a cytotoxic effect [23]. Thus, the cell culture studies clearly indicate that both L-VC and D-VC demonstrated a comparable induction of oxidative stress leading to mitochondrial membrane depolarization in Kras^G12D^-expressing cancer cells. It was evident only if both VC enantiomers were used in combination with ATO. While a natural form of VC has been tested in clinical settings as a cancer drug at a high dosage, its unnatural enantiomer D-VC, also known as erythorbic acid, is mostly utilized as a food preservative. Considering that D-VC has the same potency in inducing oxidative stress in cancer cells as L-VC, it is likely that both forms of VC have a potential in cancer therapeutics.

### 2.2. Evaluations of the Toxic Effects of D-VC and L-VC in Mice

Applications with high doses of VC have been proposed for KRAS mutant cancer treatment related to the glucose-dependent absorption of the oxidized form of VC, dehydroascorbate, by cancer cells. If both forms of VC (L-VC and D-VC) demonstrate a similar cytotoxic impact on cancer cells at a high concentration (1 mM range) when combined with ATO, it is imperative to carry out a toxicity study and to identify the less toxic form of VC for its clinical applications.

To compare the toxicity effects of L-VC and D-VC, the injections of either VC enantiomer in high doses were performed daily with the calculated dose of D-VC and L-VC at 9.2 g/kg. A considerably high dosage was toxic and it reduced the viability of mice. We observed that the effects of L-VC were more toxic compared to D-VC. On the first day after drug injections, the survival rate in the L-VC group decreased to 80%, while in the D-VC group, all mice remained alive. This indicator remained for 9 days. After 10 days of L-VC and D-VC injections, the survival rates decreased in the studied groups to 40% and 80%, respectively. Survival in the L-VC group on day 15 decreased to 20%. Until the end of the course of injections, the survival rate in the D-VC group remained at 80%, while all mice in the L-VC group died (Figure 3).

According to the VC (9.2 g/kg) injection study, the effect of L-VC had a more pronounced toxic effect on the organisms of mice compared to D-VC and led to the lethality of all mice in the group following 18 days of the drug injection course (*p* = 0.0157). Only 20% lethality was detected in the D-VC-injected mice after 18 days of the drug injections.

In addition to the low survival rate, the L-VC group experienced decreased vital signs. The mice were noted to feel unwell, have poor appetite and partial hair loss. The appearance of trophic ulcers on the back and pelvis of the mice was noted (Figure 4). Trophic ulcers appeared as destructive disorders of the epithelial and basal layers of the skin and inflammatory processes were present. The foci of tissue necrosis were observed at the scabs. These effects were not observed in mice in the D-VC-injected group.

### 2.3. The Influence of D-VC and L-VC on Leukocyte Count

The positive role of vitamin C on the immune system is known. It has been reported that VC enhances the differentiation and proliferation of B and T cells [24]. It was of interest to study the effect of high doses of D-VC and L-VC on the leukocyte count.

After 18 days of a course of injections with high doses of D-VC and L-VC, a change in the leukocyte blood count was observed. In both the D-VC- and L-VC-injected groups, there was a 3.4-fold increase in monocytes (*p* = 0.0092 and *p* = 0.0003, respectively) compared to control mice.

There was a 3.7-fold decrease in the percentage of basophils in the L-VC group compared to the control (*p* = 0.0280). In the D-VC group, no significant changes in this indicator were detected (*p* = 0.4071) (Figure 5).

There were no significant changes in the profile of neutrophils, eosinophils and lymphocytes. There was a decrease in lymphocytes in the D-VC and L-VC experimental groups compared to controls, but this was not statistically confirmed.

### 2.4. Evaluation of the Effects of High Doses of D-VC and L-VC on Internal Organs

The organ histopathology assessment was performed for all studied groups. Mice in the control group that received PBS injections showed normal, well-defined histological structures without any signs of vascular or inflammatory changes.

Mice that received L-VC injections exhibited more severe toxic organ damage compared to the D-VC group (Table 1 and Figure 6). Dystrophy was observed in the organs of mice in the L-VC group, while D-VC caused degeneration of only the myocardial tissue. Moreover, it can be noted that as a result of a high dose of L-VC, the fibrosis of lung and myocardial tissue was observed.

## 3. Discussion

A potent cytotoxic oxidative stress induced by D-VC or L-VC in combination with ATO on KRAS mutant cancer cells has been reported [17]. The main finding of the present study is that both VC enantiomers have a comparable capacity in the induction of oxidative stress in KRAS mutant cancer cells, but the D-VC injections in high doses are less toxic in mice compared to its natural form, L-VC. Considering that VC has two stereocenters, it can exist in four diastereoisomeric forms. Only the natural form of VC (L-VC) represents the most active biological (antiscorbutic) form among its diastereoisomeres. D-VC (erythorbic acid) and D-ascorbic acid possess very weak antiscorbutic activity because of their poor retention in the body and rapid excretion from the body [10]. It is evident that the comparable oxidizing impacts of L-VC and D-VC on cancer cells are not related to a biological antiscorbutic activity of VC because D-VC has only 5% of antiscorbutic activity possessed by L-VC [10,25]. It has been proposed that the observed oxidizing effects of L-VC and D-VC on cancer cells are mediated by their oxidized dehydroascorbate forms as observed and described previously [4,16,17].

A natural form of VC (L-VC) has been intensively studied not only as a critical nutrient supplement, but also as a tumor suppressing drug observed in its high dose applications, while D-VC is known as a common preservative in the food industry. At a high dose, L-VC provokes oxidative stress in cancer cells mediated by an enhanced glucose absorption mechanism [4]. Our study indicates that a non-natural enantiomer of VC (D-VC) has a similar potency to L-VC in inducing oxidative stress in the Kras^G12D^-inducible mouse cancer cell line. Both VC enantiomers, L-VC or D-VC, in combination with ATO provoke a massive ROS generation that is cytotoxic to cancer cells. The cytotoxic oxidative stress of L-VC in combination with ATO on KRAS mutant cancer cells has been previously described where D-VC with ATO was also effective in killing HCT116 (KRAS mutant) cancer cells [17]. Considering that both VC enantiomers possess a comparable capacity in the induction of oxidative stress in cancer cells, it indicates that L-VC and D-VC act by a similar glucose-dependent mechanism mediated by DHA, the oxidized form of VC [18].

According to the animal VC administration study, a high dose of D-VC (9.2 g/kg) caused less stress on the organisms, which is reflected in the much higher vitality of mice. The effect of L-VC was more toxic, which was accompanied by a pronounced stress on mice and the appearance of ulcers. The L-VC pharmacology is complex and intensively studied, and its non-natural enantiomer D-VC did not attract much attention because of its deficient biological function in anti-scorbutic activity [26]. There is no clear mechanism explaining an observed lower toxicity of D-VC, but there is an original study that indicates a relatively rapid metabolic clearance of D-VC within 24 h from mammals (guinea pigs and rats), whereas L-VC circulates for a much longer time and it is retained in the body for four days [27]. A rapid excretion in urine has been noted for D-VC and it is likely that L-VC and D-VC are filtered through the glomeruli, but only L-VC is reabsorbed by tubular cells and retained in the body. This mechanism has to be further studied and it might provide an explanation why D-VC has a low retention in mammals and is less toxic compared to L-VC.

The histopathological studies were coherent with the mouse survival data by indicating the high toxic effects on the heart and lungs in the L-VC-injected mice. It can be assumed that L-VC has an aggressive effect on the vascular system. It is known that VC in high doses increases the permeability of the vascular endothelium [28] and also suppresses angiogenesis [29,30]. Moreover, mice that received L-VC injections showed a decreased number of basophils. Basophils release platelet activating factor, which increases vascular permeability [31]. The decrease in the number of basophils may be associated with the suppressive effects of L-VC on a vascular system, but this assumption requires further study.

The increase in monocytes in the blood of mice in experimental groups can be explained by the fact that high doses of D-VC and L-VC are damaging factors, which is especially noticeable in the L-VC-injected group. In this case, the number of monocytes may be increased as a result of the formation of skin ulcers. However, no extensive damage to the skin was observed after the course of D-VC injections. The role of monocytes in tumor progression is controversial. Monocytes can participate both in maintaining the malignant process and suppressing it [32]. Recently, strategies for reprogramming monocytes for anticancer therapy are being actively studied [33].

Although we did not find changes in the levels of other leukocyte parameters, a study of the effect of high doses of D-VC and L-VC on tumors is yet to be identified. Cancer patients experience a general decrease in immunity, especially after chemotherapy. VC can speed up the restoration of the organism’s immune forces. The positive effect of VC on the proliferation of T- and B-lymphocytes, as well as on NK cells, which are able to recognize and destroy cancer cells, has been reported [34]. Currently, several clinical trials are investigating the anticancer potential of NK cells generated ex vivo [35].

The D-VC and L-VC applications in high doses in combination with arsenic trioxide induce a potent cytotoxic oxidative stress in KRAS mutant cancer cells [17] and their clinical relevance is yet to be determined by performing clinical trials. Many studies have demonstrated the therapeutic effect of L-VC in suppressing tumorigenesis, while D-VC only shows its potential and might be more advantageous than L-VC in a clinical setting. Considering the lower toxicity of D-VC and its more superior effect on KRAS mutant tumor growth compared to its natural form, L-VC, it is necessary to perform clinical studies of D-VC and determine its clinical relevance.

## 4. Materials and Methods

### 4.1. Subjects

The Kras^G12D^-inducible mouse pancreatic cancer cell line (AK192) has been provided by Dr. Haoqiang Ying (Department of Molecular and Cellular Oncology, University of Texas MD Anderson Cancer Center, Houston, TX, USA) and the doxycycline-inducible expression of the Kras^G12D^ in this cell line has been validated by immunoblotting with a specific Kras^G12D^ antibody developed by Cell Signaling Technology, Danver, MA, USA.

The in vivo study included 15 adult female CD1 mice. The mice were kept in a temperature regime of ±20–23 °C, a relative air humidity of 50–60% and a 12 h light cycle with free access to water and food. All animal experiments were carried out in accordance with international principles and the requirements of the European Convention for the Protection of Vertebrate Animals Used for Experimental or Other Scientific Purposes. The study was approved by the local ethical commission of National Laboratory Astana, protocol No. 04-2021.

### 4.2. Cell Culture and Treatment

The AK192 transgenic mouse pancreatic adenocarcinoma cell line was grown in Dulbecco’s Modified Eagle Medium/Nutrient Mixture F-12 (DMEM F-12) supplemented with 10% Fetal Bovine Serum (FBS), 2 mM l-glutamine and penicillin (100 units/mL)-streptomycin (100 μg/mL). AK192 cells incubated with doxycycline (2 μg/mL) were treated with ATO alone or different forms of VC with different treatment conditions for 24 and 48 h. The cells were seeded on a 6-well plate at 150,000 cells per well and grown for 24 h, and they were subsequently treated with single drugs (5 μM ATO, 1 mM D-VC, 1 mM L-VC) or drug combinations (5 μM ATO with 1 mM D-VC or 5 μM ATO and 1 mM L-VC).

### 4.3. Flow Cytometry

The adherent cells were collected by trypsinization at 48 h post-treatment and washed and stained with MitoSOX at 2.5 μM for 30 min at 37 °C for mitochondrial superoxide production, and they were separately double stained with MitoTracker Green/MitoTracker Deep Red (150 nM and 50 nM, respectively, for 30 min at 37 °C) for mitochondrial depolarization (all from ThermoFisher, Waltham, MA, USA) [17]. After staining, the dyes were washed, cells were resuspended at 350 μL of PBS and were kept covered in the dark on ice until analyzed. Acquisition of samples was performed on an Attune NxT Flow Cytometer (ThermoFisher) according to the manufacturer’s instructions. For both assays, 30,000 events were acquired.

### 4.4. Calculation of the Dose of D-VC and L-VC

D-VC (#856061) and L-VC (#A7506) crystalline forms were purchased from Sigma-Aldrich (Burlington, MA, USA). The preparation of the drugs was carried out according to the protocol described in the article by Wu X et al. [17].

The calculation of the administered dose was based on the toxicity of VC (LD50) equal to 11.9 g/kg [36]. Data from previous studies on the toxicity of VC in mice were also taken into account [24]. According to this, a dose of 9.2 g/kg was chosen for injections of L-VC and D-VC.

### 4.5. Experimental Design

The mice were randomly assigned to three groups: an experimental group injected with L-VC (*n* = 5), an experimental group injected with D-VC (*n* = 5) and a control group (*n* = 5). Before the experiment, the mice were weighed; their weight was ±20–25 g.

D-VC and L-VC were injected into the peritoneal cavity of mice. To minimize the risk of possible organ puncture and the induction of inflammatory processes, the mice were turned head down prior to the drug injections, allowing the abdominal organs to move and provide a relatively free space for needle insertion.

D-VC and L-VC were administered daily once a day. Animals in the control group received injections of Phosphate Buffered Saline (PBS). The course of injections continued until there was a critical decrease in the viability and survival of mice in both groups. During the course of injections, animals were constantly monitored for visible signs of the toxic effects of D-VC and L-VC. After completing the course of injections, the surviving animals and the control group were removed from the experiment by decapitation after anesthesia with isoflurane.

### 4.6. Hematology (WBC)

Blood was collected from the tail vein of each mouse after decapitation with isoflurane. Venous blood smears were prepared from each sample and fixed in 96% alcohol for one hour, then dried at room temperature for 24 h. The staining was carried out according to the Romanowsky–Giemsa method. Hematology examination of the white blood cell count (WBC) was performed using an Olympus CX41 microscope (Tokyo, Japan) and included assessment of the following parameters: band neutrophil, segmented neutrophil, eosinophil count (EOS), basophil count (BAS), lymphocyte count (LYM) and monocyte count (MON).

### 4.7. Histopathology Assessment of Tissues

The material for histological analysis was fixed in 10% neutral formalin, after which they were washed under a thin stream of running water. For dehydration, the material was passed through a series of alcohol solutions in increasing concentration (70%, 80%, 90%, 96% and 100%) for 30–40 min. After posting for alcohols, the material was placed in xylene (2 shifts of 15–20 min). To improve paraffin wax impregnation, the material was kept in a mixture of paraffin/paraffin wax in a ratio of 8:2 for 1 h in a thermostat at a temperature of 58 °C. Then, the material was poured with a mixture of paraffin wax, cooled and histological sections were made. The average slice thickness on a microtome was 2–5 μm. Sections were placed on glass slides coated with a protein solution for better fixation of the material. Before staining, the slides were subjected to dewaxing and placed first in two solutions of xylene, and then they were passed through alcohols of descending concentration (100%, 96%, 90%, 80% and 70%) for 2 min each. Histological slides were stained with hematoxylin-eosin. They were stained in hematoxylin for 10 min, then washed with tap water. Then, the slides were stained in eosin for 10 min and passed through ascending concentrations of alcohols and two portions of xylene for 2 min each. To prepare permanent slides, the material was placed in Canadian balsam and fixed with a coverslip. Stained slides were analyzed and photographed using a transmitted light trinocular microscope complete with a color digital camera (Olympus BX53, Tokyo, Japan, 2013).

### 4.8. Statistical Analysis

All statistical analyses were performed using GraphPad Prism 9 software (GraphPad Software, Inc., La Jolla, CA, USA). Data are expressed as mean ± SD. Statistical differences between groups were determined using two-way ANOVA, which was performed with Tukey’s correction for multiple comparisons. Kaplan–Meier survival curves were compared using the log-rank (Mantel–Cox) test. Values of *p* < 0.05 were considered statistically significant (* *p* < 0.05, ** *p* < 0.01 *** *p* < 0.001, **** *p* < 0.0001, ns *p* > 0.05).

## 5. Conclusions

D-VC and L-VC have shown therapeutic potential for the treatment of human cancers. High doses of D-VC and L-VC demonstrated different toxic burdens on animals. Determining the optimal strategy for combating tumors is an urgent task. We show that both VC enantiomers have a similar oxidizing capacity on AK192 (Kras^G12D^-inducible) cancer cells that is enhanced in combination with ATO to a robust ROS generation and mitochondrial membrane depolarization leading to a cytotoxic effect in cancer cells. It outlines a promising potency of the combination of VC and ATO in clinical settings. Moreover, our studies conducted on CD1 mice demonstrate that L-VC at a high dose (9.2 g/kg) has an aggressive effect on the vascular system, as well as the organs of pulmonary circulation. Severe damage to blood vessels, dystrophy and fibrosis were observed in the lung and heart myocardium. The high similar dose of D-VC did not cause substantial toxic damage to internal organs, which is coherent with the obtained mouse survival data.

We also assessed the effect of D-VC and L-VC on the leukocyte count. In general, there is a change in the level of basophils and monocytes. However, we did not find any difference between the effects of D-VC and L-VC on immune status.

Thus, the results of our studies provide new opportunities for developing an optimal therapeutic approach to the treatment of KRAS mutant cancers applying high doses of D-VC in combination with ATO.

## Figures and Tables

**Figure 1 ijms-25-02531-f001:**
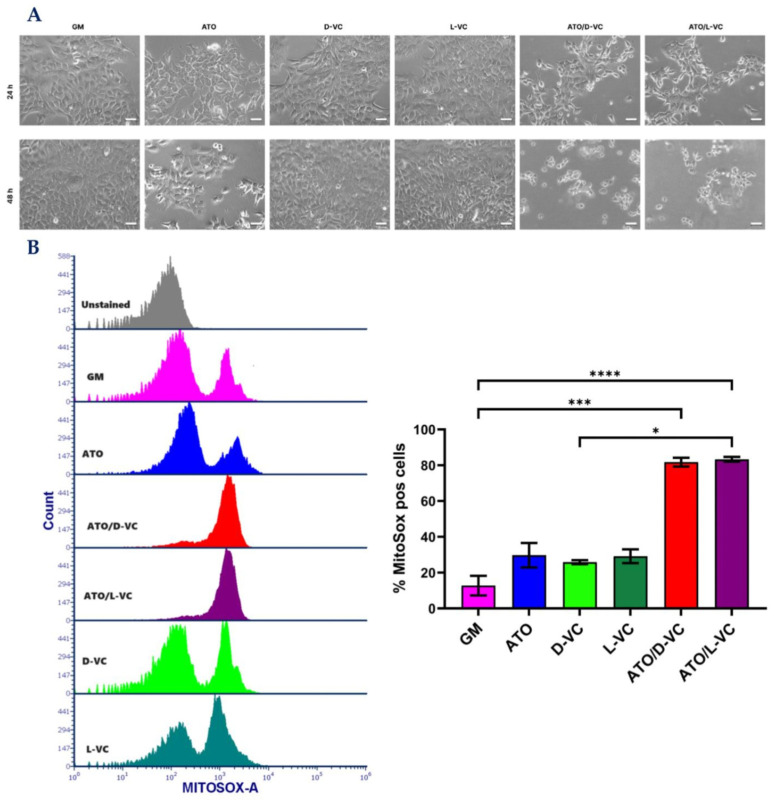
Induction of ROS generation in AK192 cancer cells by the combination of ATO with L-VC or D-VC. (**A**) Microscopical images of wells with different treatment conditions for 24 and 48 h. Scale bar, 50 µm. (**B**) A flow cytometric analysis of AK192 cancer cells treated with indicated compounds for 48 h. Cells were collected and incubated with Mitosox, staggered overlay of single experiment and bar chart of min three independent experiments (mean ± SD). * *p* < 0.05, *** *p* < 0.001, **** *p* < 0.0001.

**Figure 2 ijms-25-02531-f002:**
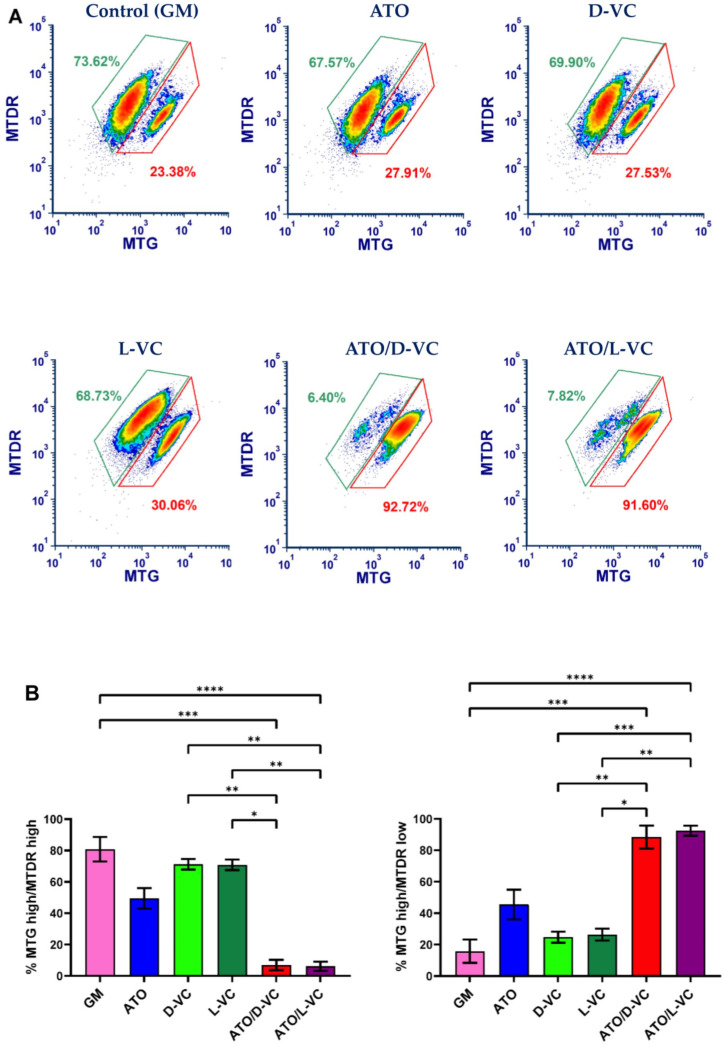
MitoTracker staining analysis of AK 192 cell line treated with 5 μM ATO alone, 1 mM D-VC, 1 mM L-VC and drug combinations of 5 μM ATO with 1 mM D-VC, 5 μM ATO and 1 mM L-VC for 48 h. (**A**) Flow cytometric analysis of MitoTrackers, example of single experiment results and (**B**) the bar chart of results representing min three independent experiments (mean ± SD). * *p* < 0.05, ** *p* < 0.01, *** *p* < 0.001, **** *p* < 0.0001.

**Figure 3 ijms-25-02531-f003:**
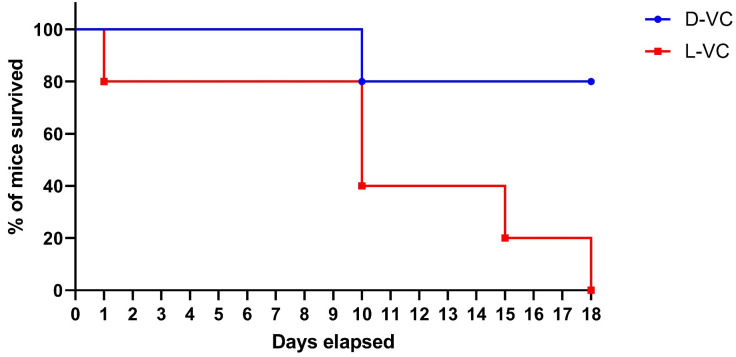
Indicators of the viability of mice during the course of injections with D-VC and L-VC (*n* = 5 in each group). Death of mice that received L-VC injections occurred on days 1, 10, 15 and 18 of the course. All mice in the L-VC group died. Death of mice that received D-VC injections occurred on day 10 of the injection course. Survival rate in the D-VC group was 80%.

**Figure 4 ijms-25-02531-f004:**
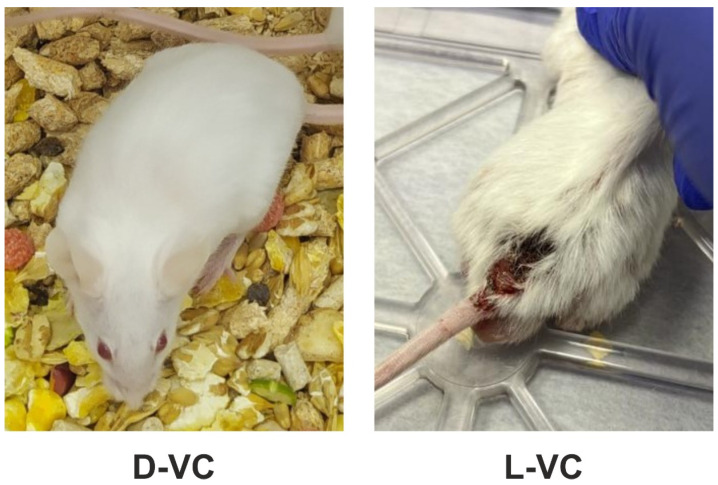
Macroscopic monitoring of mice during intraperitoneal injections of D-VC and L-VC for 18 day. The study included 15 adult female CD1 mice. The mice received intraperitoneal injections of D-VC and L-VC at a dose of 9.2 g/kg daily once a day for 18 days. Control animals received injections of PBS. The formation of trophic ulcers on the back and pelvis of the mice of L-VC injected group: damaged epithelium and basal layers of the skin, inflammatory processes, scabs and tissue necrosis.

**Figure 5 ijms-25-02531-f005:**
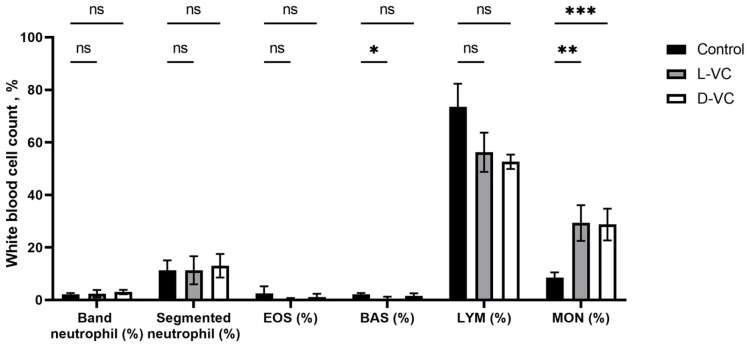
Indicators of the leukocyte blood count of mice. The study included 15 adult female CD1 mice. The mice received intraperitoneal injections of D-VC and L-VC at a dose of 9.2 g/kg daily once a day for 18 days (*n* = 5 in each group). Control animals received injections of PBS. Monocytes increased 3.4-fold in both the L-VC- and D-VC-injected groups (*p* = 0.0092 and *p* = 0.0003) as compared to control animals. The percentage of basophils in the L-VC group was 3.7 times lower than in the control group (*p* = 0.0280). This indicator showed no significant changes in the D-VC group (*p* = 0.4071) (* *p* < 0.05, ** *p* < 0.01 *** *p* < 0.001, ns *p* > 0.05).

**Figure 6 ijms-25-02531-f006:**
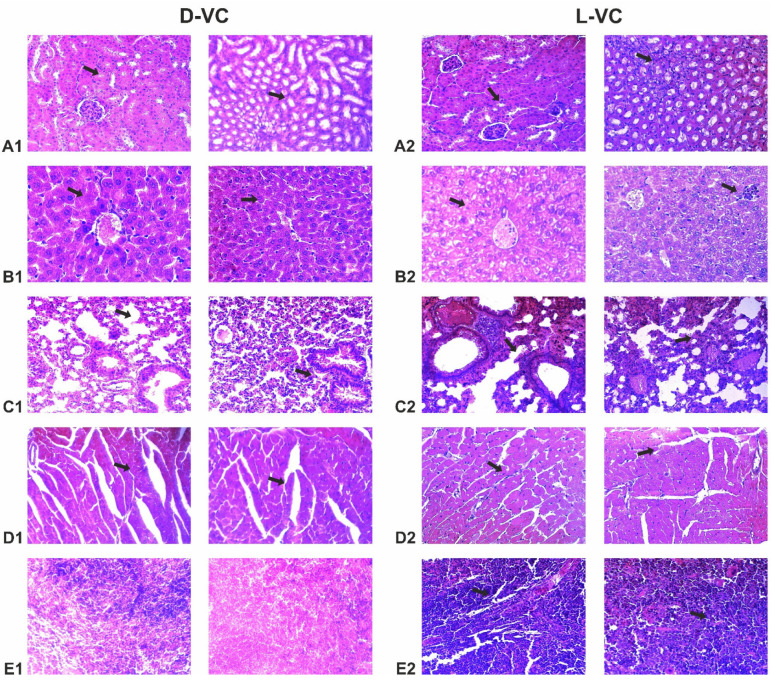
Histological analysis after intraperitoneal injections of D-VC and L-VC at a dose of 9.2 g/kg daily once a day for 18 days. The hematoxylin and eosin-stained tissue, ×200. (**A1**) Kidney. Swelling (arrow) of the stroma and dystrophic changes (arrow). (**A2**) Kidney. Dystrophy (arrow), swelling of the epithelium of the renal tubules, uneven hyperplasia of podocytes (arrow). Perevascular lymphocytic infiltration. (**B1**) Liver. Protein edema of the stroma, pronounced congestion of blood vessels (arrow). (**B2**) Liver. Congestion of blood vessels, protein degeneration of hepatocytes (arrow), lysed erythrocytes in the lumen of arterioles, single perivascular lymphocytic infiltrates (red arrow). (**C1**) Lungs. Rupture of interalveolar septa, foci of distelectasis (arrow). Erythrostasis and sludge of erythrocytes in the lumen of the areolas (arrow). (**C2**) Lungs. Acute vascular congestion with multiple confluent hemorrhages (arrow). Multiple foci of thrombosis and perivascular hemorrhages (red arrow). (**D1**) Myocardium. Protein dystrophy of cardiomyocytes (blue arrow), small foci of wave-like contracture of muscle fibers (arrow). (**D2**) Myocardium. Protein dystrophy and destruction of cardiomyocytes (red arrow). Perivascular fibrosis with lymphocytic infiltration (arrow). (**E1**) Spleen. The structure is not broken. (**E2**) Spleen. Areas of thrombosis and hemorrhage and vascular congestion (arrow).

**Table 1 ijms-25-02531-t001:** Comparative characteristics of histopathological changes after exposure to D-VC and L-VC.

Organs	D-VC	L-VC
Kidneys	Swelling of the stroma, sludged erythrocytes in the lumen of blood vessels. The epithelium of the renal tubules is swollen, with pronounced dystrophic changes, and the lumens of the tubules are narrowed.	Dystrophy, swelling of the epithelium of the renal tubules, in the center of the capillary glomerulus with symptoms of erythrodiapedesis, uneven hyperplasia of podocytes. Perevascular lymphocytic infiltration.
Liver	Protein edema of the stroma, pronounced congestion of blood vessels, single perivascular lymphocytic infiltrates.	Congestion of blood vessels, protein degeneration of hepatocytes, lysed erythrocytes in the lumen of arterioles.
Lung	Erythrostasis in the microcirculatory bed, acute vascular congestion, sludge of erythrocytes in the lumen of the areolas. Rupture of interalveolar septa, foci of distelectasis. Perivascular lymphocytic infiltrates.	Acute vascular congestion with multiple confluent hemorrhages, thrombosis of microvasculature vessels, sludged erythrocytes. Multiple foci of thrombosis and perivascular hemorrhages are signs of severe intoxication. Fibrinoid necrosis of the vascular wall is a sign of long-term destruction of the vascular walls. Signs of acute inflammation (interstitial pneumonia)—lymphocytic—plasmacytic infiltration of peribronchial tissue and interalveolar septa (interstitial pneumonia—a reaction to intoxication in the organism). Collapse of lung tissue.
Myocardium	Protein dystrophy of cardiomyocytes, small foci of wave-like contracture of muscle fibers, sludge of erythrocytes in the lumen of some vessels.	Protein dystrophy of cardiomyocytes, foci of wavy contracture of fibers, sludge of erythrocytes, foci of destruction of cardiomyocytes, proteinaceous edema of the stroma. Perivascular fibrosis with lymphocytic infiltration.
Spleen	The structure is not broken.	Splenomegaly, vascular congestion, areas of thrombosis and hemorrhage.

## Data Availability

The data used to support the findings of this study are available upon request from the corresponding authors.

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
