# Peer review of "The Vitamin C Enantiomers Possess a Comparable Potency in the Induction of Oxidative Stress in Cancer Cells but Differ in Their Toxicity"

_ijms, 2024, doi:10.3390/ijms25052531_

Round 1

Reviewer 1 Report

Comments and Suggestions for Authors

In this work, authors in vitro studied the potential of VC in combination with As2O3 for the induction of oxidative stress in KrasG12D expressing mutant cancer cells as well as by in vitro study they analyzed the toxicity of enantiomers L-VC and D-VC.

I consider the work interesting but I identified some relevant problems:

a) this work has 2 different parts (in vitro vs in vivo studies) and in portions of the text they seemed relatively independent – therefore, in my opinion, authors must clarify the liaison between these two parts; in addition it is difficult to relate the concentrations studied in in vitro and in vivo studies – I consider this point important because the effects can be concentration-dependent…

b) Abbreviations must be reviewed in all manuscript. Examples: D-VC and L-VC in title, ROS in abstract, others

c) I consider that important reviews on vitamin C were not referenced in the manuscript. Examples: https://pubmed.ncbi.nlm.nih.gov/33668681/ ; https://pubmed.ncbi.nlm.nih.gov/34942996/

d) As ascorbic acid has 2 stereocenters, it can exist as 4 diastereoisomeric forms. In addition, some authors (e.g. https://pubmed.ncbi.nlm.nih.gov/33668681/) consider that at least L‐ascorbic acid, erythorbic acid, D‐ascorbic acid have vitamin C activity. Therefore, the sentence “VC exists in two stereoisomeric forms: the L- and D-forms (D-VC and L-VC)” must be improved, considering the points indicated

e) this work is a continuation of a previous one (https://onlinelibrary.wiley.com/doi/full/10.1002/ijc.32658 ) - I consider that it is important to clarify what was studied in the previous work and the novelty in the present work, concerning in vitro studies; in addition, authors must clarify the differences in bioactivity and pharmacokinetics of ALL 4 diastereoisomeric forms of L-VC, not only in cancer but in other biological effects

f) Authors referred “4.4. MTS Assay” but I could not find the corresponding results and discussion

Comments on the Quality of English Language

Minor editing of English language required

Author Response

Dear Reviewer,

We greatly appreciate You for their constructive criticism and support for our study. Your comments and suggestions have improved the manuscript substantially.

The point-by-point response to the reviewer 1 comments:

  1.      Delete “: D-VC is less toxic than its natural form L-VC” from the title.

The title has been edited as recommended by deleting the wording “D-VC is less toxic than its natural form L-VC”.

  1.      The combinations of L- and D-VC with arsenic trioxide (ATO) were tested in vitro but not in vivo, and thus the authors need to explain why they evaluated the mouse toxicity for L- and D-VC only. The authors may need to test the mouse toxicity of the combinations of L- and D-VC with ATO.

The Reviewer’s comment makes good sense. The combinations of L-VC and D-VC with arsenic trioxide (ATO) was examined on cancer cells to show that both Vitamin C enantiomers demonstrate a similar potency in killing cancer cells by inducing cytotoxic oxidative stress in combination with ATO. Considering that D-VC has a higher potency in suppressing KRAS mutant xenograft tumors (the previous published study by the authors, Wu et al.), the present study is focused on studying the toxicity of D-VC. ATO is a well-studied poisonous oxidative compound and it is not a focus of this study. The presented manuscript is a first original study that clearly demonstrates a lower toxicity of D-VC compared to its natural enantiomer L-VC that sets an initial step in promoting D-VC to its first and highly demanding clinical application.

  1.  Check the fonts for 4.3 and 4.4 sections, which seem double lined.

The font for the indicated sections has been corrected.

Reviewer 2 Report

Comments and Suggestions for Authors

Journal: International Journal of Molecular Sciences
Manuscript ID: ijms-2819615

Manuscript Type: Article

Title: The vitamin C enantiomers possess a comparable potency in induction of
oxidative stress in cancer cells but differ in their toxicity: D-VC is less toxic than its natural form L-VC

Authors: Dinara Begimbetova, Agata N. Burska, Aidana Baltabekova, Assiya Kussainova, Assiya Kukanova, Fatima Fazyl, Milana Ibragimova, Kenzhekyz Manekenova, Abay Makishev, Rakhmetkazhi I. Bersimbaev, and Dos D. Sarbassov

In the present study, the combinations of L- and D-VC with arsenic trioxide (ATO) were found to induced cytotoxic oxidative stress in Kirsten rat sarcoma (KRAS) G12D expressing mutant cancer cells. However, the natural form of VC (L-VC) is more toxic than D-VC when tested in vivo, indicating that these two VC enantiomers show a similar potency in induction of an oxidative stress in cancer cells, but D-VC has a distinctive lower toxicity to mice, when compared to L-VC. This manuscript could be published in International Journal of Molecular Sciences after the authors revise it following the suggestions shown below.

1.      Delete “: D-VC is less toxic than its natural form L-VC” from the title.

2.      The combinations of L- and D-VC with arsenic trioxide (ATO) were tested in vitro but not in vivo, and thus the authors need to explain why they evaluated the mouse toxicity for L- and D-VC only. The authors may need to test the mouse toxicity of the combinations of L- and D-VC with ATO.

3.      Check the fonts for 4.3 and 4.4 sections, which seem double lined.

Author Response

Dear reviewer,

We greatly appreciate You for their constructive criticism and support for our study. Your comments and suggestions have improved the manuscript substantially.

The point-by-point response to the reviewer 2 comments:

a) this work has 2 different parts (in vitro vs in vivo studies) and in portions of the text they seemed relatively independent – therefore, in my opinion, authors must clarify the liaison between these two parts; in addition it is difficult to relate the concentrations studied in in vitro and in vivo studies – I consider this point important because the effects can be concentration-dependent…

The main reason of the cell culture studies (in vitro part) is to demonstrate that L-VC and D-VC have a similar potency in inducing oxidative stress associated with mitochondria membrane depolarization. The most critical part of the manuscript is the mouse toxicity study (in vivo study) that shows for the first time that D-VC is less toxic compared to its natural enantiomer L-VC. The presented manuscript is establishing platform for introducing D-VC for its clinical applications that appeared to be less toxic than L-VC.

b) Abbreviations must be reviewed in all manuscript. Examples: D-VC and L-VC in title, ROS in abstract, others

As recommended D-VC and L-VC were deleted from the title. The “ROS” acronym has been spelled out in the Abstract.

c) I consider that important reviews on vitamin C were not referenced in the manuscript. Examples: https://pubmed.ncbi.nlm.nih.gov/33668681/; https://pubmed.ncbi.nlm.nih.gov/34942996/

As recommended both important reviews were referenced in the edited manuscript as the references #10 and #11.

d) As ascorbic acid has 2 stereocenters, it can exist as 4 diastereoisomeric forms. In addition, some authors (e.g. https://pubmed.ncbi.nlm.nih.gov/33668681/) consider that at least L‐ascorbic acid, erythorbic acid, D‐ascorbic acid have vitamin C activity. Therefore, the sentence “VC exists in two stereoisomeric forms: the L- and D-forms (D-VC and L-VC)” must be improved, considering the points indicated

We fully agree with the Reviewer and the sentence was edited as indicated: “The two enantiomers of VC known as the L- and D-forms of VC or ascorbic acid (D-VC and L-VC) have a similar oxidizing effect on cancer cells…”. (the text lines 84-85).

e) this work is a continuation of a previous one (https://onlinelibrary.wiley.com/doi/full/10.1002/ijc.32658) - I consider that it is important to clarify what was studied in the previous work and the novelty in the present work, concerning in vitro studies; in addition, authors must clarify the differences in bioactivity and pharmacokinetics of ALL 4 diastereoisomeric forms of L-VC, not only in cancer but in other biological effects

The Reviewer point is well taken. We would like to clarify that in our previous study, we show that in cell culture both D-VC and L-VC in combination with ATO have a similar cytotoxic oxidative stress in HCT116 (KRAS mutant cancer cells) as indicated by apoptotic assay and also ROS detection. In the present study, we expand this finding further by studying impact of ATO/D-VC or ATO/L-VC combination on AK192 (KrasG12D mutant inducible expression cell line) cancer cell line by performing ROS detection and examining the mitochondrial membrane potential study. Most importantly, we show for the first time that D-VC is less toxic than its natural enantiomer L-VC. It is a critical finding defining the path for promising clinical applications of D-VC in cancer treatments.

The reviewer makes an important point that the VC molecule contains two chiral carbon atoms, C4 and C5. It means VC can exist in 4 diastereoisomeric forms but a natural form of VC has only one enantiomer D-VC (or also known as d-isoascorbic acid) that has only 5% of VC anti-scurvy activity. The focus of the presented manuscript is D-VC, the only enantiomer of VC, the other possible stereoisomers of VC has not been studied or discussed. Our main interest is the clinical applications of VC and only L-VC and D-VC are FDA approved compounds among four VC stereoisomers. D-VC (d-isoascorbic acid) is mostly used as an antioxidant additive in processed foods.  Considering our study indicating a lower toxicity of D-VC compared to L-VC, D-VC is becoming very attractive for clinical applications.

f) Authors referred “4. MTS Assay” but I could not find the corresponding results and discussion

We are very thankful to the Reviewer for identifying the problem. The MTS assay has not been performed and the section 4.4 has been deleted. 

Round 2

Reviewer 1 Report

Comments and Suggestions for Authors

Authors only partially answered the points raised previously:

a) this work has 2 different parts (in vitro vs in vivo studies) and in portions of the text they seemed relatively independent – therefore, in my opinion, authors must better clarify the liaison between these two parts; in addition it is difficult to relate the concentrations studied in in vitro and in vivo studies – I consider this point important because the effects can be concentration-dependent…

b) Abbreviations must be reviewed in all manuscript. Examples: GTPase, OTHERS

c) As ascorbic acid has 2 stereocenters, it can exist as 4 diastereoisomeric forms. In addition, some authors (e.g. https://pubmed.ncbi.nlm.nih.gov/33668681/) consider that at least L‐ascorbic acid, erythorbic acid, D‐ascorbic acid have vitamin C activity. Therefore, this point must be developed and clarified by authors in the manuscript; in addition, authors must clarify the differences in bioactivity and pharmacokinetics of ALL 4 diastereoisomeric forms of L-VC, not only in cancer but in other biological effects

Comments on the Quality of English Language

Minor editing of English language required

Author Response

Response to Reviewer 1 Comments

Please see the attachment also

1. Summary

We greatly appreciate both reviewers for their constructive criticism and support for our study. The reviewers’ comments and suggestions have improved the manuscript substantially.

2. Questions for General Evaluation

Reviewer’s Evaluation

Response and Revisions

Does the introduction provide sufficient background and include all relevant references?

Yes

Please see comments below

Are all the cited references relevant to the research?

Yes

Is the research design appropriate?

Yes

Are the methods adequately described?

Yes

Are the results clearly presented?

Yes

Are the conclusions supported by the results?

Yes

3. Point-by-point response to Comments and Suggestions for Authors

Comments: a) this work has 2 different parts (in vitro vs in vivo studies) and in portions of the text they seemed relatively independent – therefore, in my opinion, authors must better clarify the liaison between these two parts;

Response a: We apologize that the manuscript did not clarify why it has two (in vitro and in vivo) parts that seem independent. The main goal of the paper is to show that D-VC (D-form of Vitamin C) is less toxic than its natural form L-VC. It is a critical feature considering that we are working on D-VC as a component of the anti-cancer drug combination. We show that D-VC and L-VC have the same potency in killing KRASG12D oncogene expressing cancer cells by inducing a cytotoxic oxidative stress (Fig. 1) associated with a depolarization of mitochondria (Fig. 2). It is clear that D-VC and L-VC are effective only in combination with arsenic trioxide (ATO) and they are not effective on their own. Besides, it has been shown for the first time that the ATO/L-VC or ATO/D-VC combination acts by inducing depolarization of mitochondrial membrane.

As recommended by the Reviewer, we connected the in vivo (the mouse toxicity study) by introducing the following sentences (in lanes 177-184):

Thus, the cell culture studies clearly indicate that both L-VC and D-VC demonstrated a comparable induction of an oxidative stress leading to mitochondrial membrane depolarization in KrasG12D expressing cancer cells. It has been evident only if both VC enantiomers were acted in combination with ATO. While a natural form of VC has been tested in clinical settings as anti-cancer drug in high dosage, its unnatural enantiomer D-VC also known as erythorbic acid is mostly utilized as a food preservative. Considering that D-VC has a same potency in inducing oxidative stress in cancer cells as L-VC, it is likely that both forms of VC have a potential in cancer treatment.

a) (the comment continuation) in addition it is difficult to relate the concentrations studied in in vitro and in vivo studies – I consider this point important because the effects can be concentration-dependent…

The Reviewer’s point makes a good sense. The comment is well taken and we apologize for the confusion and provide the following clarification. D-VC has never been utilized as a drug previously. It is critical to assess its toxicity by comparing it with a natural form of VC (L-VC). The purpose of in vivo study was carried out to determine which form of VC (L-VC or D-VC) is less toxic in mice. So, there was no purpose to relate a dose of VC effectively killing cancer cells in combination with ATO that was demonstrated in vitro (the cell culture study). Our study points out that D-VC is less toxic compared to L-VC that defines the advantage of D-VC over L-VC.

The following sentences were incorporated to clarify the main purpose of the toxicity studies (lanes 188-194):

The high doses of VC applications have been proposed for KRAS mutant cancer treatment related to a glucose-dependent absorption of the oxidized form of VC dehydroascorbate (in a form of bicyclic hemiketal) by cancer cells. If both form of VC (L-VC and D-VC) demonstrate a similar cytotoxic impact on cancer cells in high concentration (1 mM range) when combined with ATO, it is imperative to carry out the toxicity study and to identify a least toxic form of VC for its clinical applications. To compare the toxicity effects of L-VC and D-VC, the injections of both VC components in high dose…

Comments: b) Abbreviations must be reviewed in all manuscript. Examples: GTPase, OTHERS

Response b: As advised by the Reviewer, the abbreviations were reviewed and their full names were properly placed.

Comments: c) As ascorbic acid has 2 stereocenters, it can exist as 4 diastereoisomeric forms. In addition, some authors (e.g. https://pubmed.ncbi.nlm.nih.gov/33668681/) consider that at least L‐ascorbic acid, erythorbic acid, D‐ascorbic acid have vitamin C activity. Therefore, this point must be developed and clarified by authors in the manuscript; in addition, authors must clarify the differences in bioactivity and pharmacokinetics of ALL 4 diastereoisomeric forms of L-VC, not only in cancer but in other biological effects

Response c: We are thankful to the Reviewer who raised the point that was missed in our manuscript. In our study, we focused on the oxidation effect of VC on KRAS mutant cancer cells mediated by the oxidized form of VC dehydrosacorbate. The main anti-scurvy activity of VC has not been discussed in the manuscript. As recommended by the Reviewer, we introduced a few sentences in the introduction section (Lanes 68-75).

VC is an essential nutrient for overall health based on its electron-donating property and it functions as an enzymatic cofactor in a variety of reducing reactions within deoxygenases (involved in synthesis of collagen and carnitine) and monooxygenases (required for synthesis of hormones). VC is defined as a critical anti-oxidant because it is not made in human cells and its deficiency leads to scurvy. Besides, VC’s functions have been also linked to the fundamental cellular processes including transcription and translation via different mechanisms (Ref. new). Recently, the high dose VC applications became an attractive approach in cancer therapy…

To address the bioactivity and and pharmacokinetics of ALL 4 diastereoisomeric forms of L-VC, not only in cancer but in other biological effects, we included the following sentences in the Discussion section (Lanes 294-303):

Considering that VC has two stereocenters, it can exist in four diastereoisomeric forms. Only a natural form of VC (L-VC) represents a most active biological (antiscorbutic) form among of its diastereoisomeres. D-VC (erythorbic acid) and D‐ascorbic acid possess a very weak antiscorbutic activity because of their poor retention in the body and a rapid excretion from the body. It is evident that the comparable oxidizing impacts of L-VC and D-VC on cancer cells is not related to a biological antiscorbutic activity of VC because D-VC has only 5% of antiscorbutic activity possessed by L-VC (Ref. PMID: 7446456, by Goldman, 1981). It has been proposed that the observed oxidizing effects of L-VC and D-VC on cancer cells is mediated by their oxidized dehydroascorbate forms as observed and described previously [4, 16, 17].

Response 1: The editing of manuscript (English language) has been completed.

Round 3

Reviewer 1 Report

Comments and Suggestions for Authors

The text was improved.